# Fast Terahertz Reflection Imaging for In-Line Detection of Delaminations in Glass Fiber-Reinforced Polymers

**DOI:** 10.3390/s25030851

**Published:** 2025-01-30

**Authors:** Peter Fosodeder, Michael Pfleger, Kausar Rahman, Tom Dutton, Sophie Cozien-Cazuc, Sandrine van Frank, Christian Rankl

**Affiliations:** 1Research Center for Non-Destructive Testing (RECENDT) GmbH, Altenberger Strasse 66a, 4040 Linz, Austriamichael.pfleger@recendt.at (M.P.);; 2Far-UK Ltd., Unit 29, Wildford Industrial Estate, Ruddington Lane, Nottingham NG11 7EP, UKtom.dutton@far-uk.com (T.D.);

**Keywords:** terahertz imaging, image processing, robot-guided NDT, GFRP, composite materials, in-line monitoring, THz-TDS

## Abstract

Terahertz (THz) is an emerging technology particularly well suited for the non-destructive investigation of inner structures in polymers. To realize its full potential, THz imaging systems adapted to industrial constraints as well as more application studies in areas of interest are needed. In this work, we present a fast and flexible THz imaging system comprising hardware and software and demonstrate its capabilities for the investigation of defects in glass fiber-reinforced polymers (GFRPs), particularly for the detection of drilling-induced delaminations. Measurement data obtained by raster scanning of GFRP samples are gathered in 3D volumetric images. THz images of the drilled holes are then compared to reference images of the same holes obtained from X-ray computed tomography measurements. We show that THz imaging is capable of identifying not only artificial defects in the form of aluminum and Teflon inlays, but also real defects such as delaminations generated by drilling operations, and is suitable for non-destructive testing in industrial conditions.

## 1. Introduction

Fiber-reinforced composites are crucial materials for the manufacture of future generation structures such as lighter aircraft or wind turbines. Their high stiffness-to-density and strength-to-weight ratios, among other advantageous properties, make them an attractive alternative to traditional materials. However, defects can arise at all stages of the composite lifecycle, including early on during manufacturing, which can damage the structure of the material and affect its performance. Hole drilling in particular is a very common type of machining, as composite laminates often need to be joined to other materials by rivets or bolts. However, the drilling of fiber-reinforced polymers is a complex process which differs significantly from the machining of conventional metals and alloys due to the anisotropic, non-homogeneous, highly abrasive, and hard reinforced fibers characteristic of these materials. Several undesirable damages such as delaminations and fiber pull-out induced by drilling drastically reduce strength against fatigue, thus degrading the long-term performance of composite laminates [1]. Among the problems caused by drilling, delamination, a type of defect where two or more layers become separated, is considered the major type of damage [2].

Efficient quality control of the structural integrity of fiber-reinforced composites is therefore crucial for safety and reliability. The current established methods of non-destructive testing (NDT) include visual inspection [3], ultrasound imaging [4], and X-ray tomography [5]. While visual inspection can be fast, easily implementable, and cost effective, it is limited to surface inspection. Ultrasonic imaging, on the other hand, is a volumetric imaging procedure that also allows for detecting defects hidden below the surface. However, it requires physical contact with the sample and has only limited sensitivity in the direct vicinity of the surface, where delaminations are most likely to occur. Finally, X-ray computed tomography provides the best resolution across the entire volume of the sample with the potential to resolve even individual fibers. However, it uses ionizing radiation and is thus harmful to humans. This is a major drawback because the necessity for radiation shielding restricts the inline capability and significantly increases the cost.

An emerging alternative, the already mentioned method for the fast, robust, and contactless inline testing of glass fiber-reinforced polymer (GFRP) parts, is Terahertz (THz) imaging. The low-energy THz radiation is non-ionizing and therefore harmless to human operators working nearby, making it much more cost effective and better suited for inline inspection than X-ray imaging. While it does not allow for inspecting individual fibers in the material, it is very well capable of identifying regions with varying optical material parameters, such as delaminations. In contrast to ultrasound imaging, the optical properties of THz radiation allow for high sensitivity near the surface, while achieving a relatively high penetration depth in GFRP. Several groups have already conducted research in this field using artificial defects, e.g., Teflon foils embedded in a GFRP matrix, to simulate delamination [6]. To date, there is very limited literature on the study of actual defects occurring in real-life scenarios.

Meanwhile, the recent developments of high-speed, fiber-based, compact THz time-domain spectrometers (THz-TDS) [7,8] are making this technology more accessible for practical applications. THz imaging is already in use in the automotive industry [9,10], the medical field [11], security [12,13], and the polymer industry [14,15,16,17]. High-speed THz-TDS has the potential to significantly increase the range of application, on the condition that suitable scanning tools are also available. A new type of system is especially suited for practical application, the ECOPS (Electronically Controlled Optical Sampling) THz-TDS system TeraFlash smart developed by TOPTICA Photonics AG (Munich, Germany). Its key advantage is its measurement speed of 1.6 kHz, much higher than that of systems using a mechanical delay stage (around 100 Hz) or even the fast ASOPS (Asynchronous Optical Sampling) systems (up to 1 kHz).

However, to make use of these high speeds, the previously available imaging solution is not suitable. This is why we develop an integrated THz measurement platform based on a state-of-the-art THz-TDS system which allows fast spatially resolved reflection imaging for measurement automation, data processing, and visualization with a custom-made user-friendly software. In this paper, we describe this platform and demonstrate the system’s capabilities for the fast structural analysis of GFRP parts and, in particular, its use for the detection of processing-induced defects around holes in drilled GFRP laminates. We perform THz measurements on different GFRP samples and develop specialized algorithms for the analysis of the THz data. Initial evaluations are performed using simulated defects in GFRP laminates, followed by measurements on samples with real defects caused by drilling. The results are compared with reference images of the samples obtained with X-ray computed tomography.

## 2. Materials and Methods

### 2.1. Manufacturing of GFRP Samples

The GFRP plates shown in Figure 1 were manufactured by FAR UK from VTC401-G870-32%RW-1250 mm epoxy component prepreg laminates distributed by the company SHD composites (Sleaford, UK). These laminates were woven in a 2 × 2 twill pattern and stacked with an alternating 0°/90° orientation. Finally, the system was cured by vacuum bagging at a linearly increasing temperature from room temperature up to 120 °C (ramp rate 3 °C/min). The temperature of 120 °C was then held for an additional 45 min. Following this process, two plates with 3 mm and 6 mm thickness were fabricated for drilling. An additional 3 mm thick GFRP plate was produced that includes inlays made of aluminum and PTFE at different depths.

One of the challenges of the drilling process is the high amount of torque required to overcome the friction between the drill bit and the laminate. Furthermore, these strong mechanical forces cause drill bits and the surrounding material to overheat quickly [18]. The increased wear and tear of the drill bit is the consequence since the cutting edge of an overheated drill bit will wear away much faster [19]. In turn, a dull cutting edge will inflict more damage to the GFRP part and generate even more friction heat. Therefore, special attention was brought to the choice of the drill bit, feed rate, and rotational frequency to ensure the efficient drilling of GFRP. An industrial robot (UR10e cobot) equipped with a 1400 W CNC spindle and a 8 mm diameter HSSM2 drill bit was used by FAR UK. By monitoring the temperature during the drilling process with an infrared camera, an optimal feed rate of 2.5 mm/s was estimated. As indicated by the dashed boxes in Figure 1, a number of holes were drilled with the same drilling bit before it was exchanged. By visual inspection of the drilled holes, one can already observe the decreasing quality of the holes with an increasing number of drilled holes for the same drilling bit. Furthermore, it can be observed that drilling the 6 mm thick plate in Figure 1b is much more challenging than drilling the 3 mm plate in Figure 1a.

### 2.2. Principles of THz Reflection Imaging

THz reflection imaging exploits the reflections of THz radiation at interfaces between different materials or media to reconstruct the geometry of a sample in 3D. These reflections are caused by step-wise transitions in optical properties (i.e., the complex refractive index or dielectric parameter) as the radiation propagates through the sample, in the same way that transitions in acoustic material parameters (i.e., the acoustic impedance) are used in ultrasound images. The imaging scheme is represented in Figure 2.

In THz-TDS [20], a short electromagnetic pulse of THz radiation is emitted by a photoconductive antenna (PCA) [21]. The direct, coherent detection of the time-domain electric field is an important feature of a THz-TDS system. It allows for evaluating the time-of-flight of a THz pulse and calculating its THz frequency spectrum (Fourier transform [22]). Guided by a system of parabolic mirrors or optical lenses, the THz pulse is focused onto the sample. The time-domain waveform reflected from the sample is guided back to the detector, where it is measured. Understanding the interaction between radiation and material is crucial to interpret the measured signals.

In general, electromagnetic wave propagation is described by a boundary value problem using Maxwell’s equations [23]. In order to illustrate the physics involved with the propagation of a THz pulse in a dielectric material, a time-dependent solution of this problem was simulated using the finite-difference time-domain method [24]. For simplicity, we used the case of a sample invariant along the out-of-plane direction for which the system can be simulated in two dimensions only [25]. Figure 3a shows a focused THz pulse [26] propagating towards the focal point $\left(x=0,\;y=0\right)$ at an incidence angle of 8°. The darker green area represents the sample with a small air gap below the surface. In Figure 3b, the simulation space at a later time is shown. Here, several reflected THz pulses are clearly visible in free space while another pulse is still propagating inside the sample. The reflected pulses originate from the top surface of the sample and the interfaces of the air gap. As shown by the simulation, the amplitude of the reflected pulses decreases with an increasing depth of the material interface due to the absorption inside the material. Furthermore, the frequency-dependent absorption coefficient [27] of GFRP broadens the pulse in space. Ultimately, this reduces the bandwidth of the reflected THz pulse for larger depths. Therefore, it is expected that delaminations close to the surface can be resolved with higher precision than deeper features.

As shown by the simulation, a high amplitude pulse in the measured signal indicates the presence of a material transition with a significant difference in optical properties and the time delay of the pulse represents the depth at which the material transition is located. Therefore, multiple measurement signals recorded from different positions along the surface of a GFRP plate can be used to create a 3D representation of the inner sample structure.

### 2.3. Fast THz Imaging Setup

The core component of the THz imaging setup implemented for this work is the commercially available THz-TDS system TeraFlash smart [8,28] manufactured by TOPTICA Photonics AG (Munich, Germany). This system contains two synchronized 1550 nm femtosecond lasers used to trigger two PCAs [7] and outputs a very short pulsed THz time signal thanks to its broad bandwidth (almost 5 THz). It is particularly well suited for industrial applications, as it achieves high measurement rates of up to 1600 time-domain signals per second and is relatively insensitive to vibrations. Moreover, the PCAs are coupled to the lasers by optical fibers and thus can be integrated in a compact, mobile, and eye-safe measurement head together with optical elements for focusing the THz beam and a cover for dusty environments.

In the configuration applied in this work, four off-axis parabolic mirrors (OPMs) were used to focus the THz beam into the sample at an incidence angle of 8°. By choosing the focal length of the OPMs accordingly, the focal spot size and the working distance between the optical parts and the sample surface could be varied. A spot size measurement according to the knife edge method [29] shows that a focal spot diameter of 1 mm could be achieved using an OPM focal length of 101.6 mm. The measurement head was also purchased from TOPTICA Photonics AG as an accessory part for the TeraFlash smart system.

A key requirement for our imaging system is the ability to scan large samples in a fast and accurate way. Therefore, an imaging platform was designed and built to move the measurement head using two motorized linear axes with encoders. The measurement head was displaced continuously over the sample according to the scanning trajectory indicated in Figure 4a, and its position was recorded synchronously with the acquisition of the THz pulses. This implementation allowed for directly calculating accurate THz images without the need for reference markers in the sample [30].

Additionally, software for controlling the measurement parameters, processing and visualizing the acquired data was developed. A screenshot of the graphical user interface (GUI) taken during a live measurement of a drilled GFRP sample is shown in Figure 4b. It allowed to control the axes positions and other scan parameters, while showing in real time a continuous amplitude scan (A-scan), a line scan (B-scan), and an area scan (C-scan) of the running measurement, as well as the current THz time-domain signal and corresponding frequency spectrum (tab not visible in the screenshot). Although entirely developed by RECENDT, the final implementation of the software was designed in collaboration with TOPTICA Photonics AG to be made commercially available, either together with their existing imaging extension or with our imaging hardware and data reading and processing routines.

### 2.4. The 3D Image Reconstruction Principle

A simplified description of the imaging process discussed in Section 2.2 can be achieved using geometrical optics. In Figure 5a, a ray (dashed arrow) propagating into a delamination (dark region) near a drilled hole is shown. At each material interface intersecting with the path of the ray, a pulse is reflected back (blue arrows) towards the detector. An example of the THz time-domain signal (see Figure 5b) showing exactly these three reflections was measured from an artificially damaged GFRP plate. Since the depth of a measured feature is encoded in the time-axis of the measured THz signal, a relation between the time-delay Δt and the depth is needed. In optical imaging, this is commonly performed by using the optical depth Λ(1)$$\Delta t=\frac{2}{c_{0}}\Lambda \;\;,$$
where $c_{0}$ is the vacuum speed of light. In a homogeneous medium, the optical depth is related to the actual geometrical depth, and the refractive index *n* by Λ=nd.

Using a large set of THz time-domain signals measured at different positions, a series of THz images in the xy-plane at different optical depths Λ, commonly referred to as C-scans, are calculated. Stacked together, these images are gathered in a 3D data-cube of spatially resolved THz data. This is achieved by the signal processing steps shown in Figure 6.

First, a digital finite impulse response (FIR) filter [31] is applied to each raw THz signal. A typical cut-off frequency used for the measurements presented below is $f_{C}=2.5\;\mathrm{THz}$ and the according pass-band frequency is $f_{P}=3.0\;\mathrm{THz}$. Furthermore, a moving average over five consecutively acquired THz signals is applied. This effectively results in spatial low-pass filtering.

As discussed in Section 2.3, the measured THz signal represents the electric field. The actual pulse waveform strongly depends on the PCA and the optical setup and typically contains multiple positive and negative oscillations per pulse (c.f. Figure 5b). Therefore, it is difficult to distinguish between reflected pulses and oscillations that are naturally included in the individual reflections. One common approach for identifying reflections is to measure a reference signal reflected from a metal plate and calculating the deconvolution of the measured signal and the reference signal. By finding the maxima in the deconvolved signal, reflected pulses can be identified [32]. However, this approach is limited to the imaging of weakly dispersive materials only. The alteration of the pulse shape caused by the strong dispersion of GFRP can lead to systematic inaccuracies. Therefore, the envelope function of the measured signal is calculated and used as a measure for the reflectivity of the material transition. This is performed by applying the Hilbert transform [33] to the pre-processed signals as shown in Figure 5b.

Next, a 3D data cube is calculated from the irregularly positioned envelope signals. As discussed in Section 2.3, the measurement data are acquired during a continuous movement of the sample on arbitrary positions $\left(x,y\right)$ along the scanning trajectory (see Figure 4). This is different from a slow stop and measure approach, where measurement data are already acquired on a regularly spaced grid. Therefore, a regular 3D (x−y−Λ) grid is specified and the measurement data are interpolated on the grid positions. Note that the Λ-axis is the optical depth of the THz signals, which is obtained from the time axis of the THz signals and the simple transformation in Equation (Equation 1). Since the time axis is equidistantly spaced and equal for each measured signal, the Λ axis is also equidistantly spaced and requires no interpolation. Therefore, the 3D interpolation can be split up into a series of 2D interpolation steps for each C-scan. This greatly reduces the complexity of this computational task.

Due to imperfect sample alignment, it is possible that the sample is slightly tilted in the 3D dataset. By fitting a tilted plane into the measured surface of the sample, the tilt can be calculated and corrected by rotating the dataset such that the top surface of the sample lies in the xy-plane. This step is optional, but it simplifies the data visualization because each C-scan is then parallel to the surface (i.e., parallel to the glass fiber plies in the GFRP part).

Note that in practice, the reflection measurement is performed at an incidence angle of 8° (see Section 2.3), rather than at a normal incidence. In general, this leads to refraction at each non-perpendicular material interface in the sample, which is known to produce systematic errors in THz imaging [34]. As performed in our related work on THz transmission imaging [16,17], this can be accounted for by using a priori knowledge about the nominal sample geometry in order to calculate a more accurate propagation path of the ray. Such an approach is currently under development for reflection imaging. In this work, however, the systematic error is neglected due to the very small incidence angle.

### 2.5. X-Ray Computed Tomography Setup

In order to assess the results obtained with our THz reflection imaging system, X-ray computed tomography (CT) measurements were performed. These measurements were performed by the research group for computed tomography at the University of applied sciences Upper Austria - Research & Development LTD using the commercially available micro CT scanning system RayScan 250E from RayScan Technologies GmbH (Meersburg, Germany). For each sample, a 3D CT image was calculated from 1350 projections with a voxel size of 20 µm. The X-ray tube was powered by a source voltage of 180 kV, and the detector integration time was set to 666 ms.

## 3. Results

In this section, a measurement performed on a sample including artificial defects in the form of inlays of different types and sizes is shown in order to demonstrate the capabilities of the developed approach. Furthermore, the possibility of identifying defects in drilled GFRP plates is discussed based on the results of a variety of THz scans of holes in GFRP samples. The same holes were scanned using X-ray CT to allow for a direct comparison between the 3D THz dataset and the 3D X-ray CT dataset.

### 3.1. Imaging of Artificial Defects in GFRP

Figure 7a shows a sketch of the sample including the aluminum and PTFE inlays marked as dark and bright rectangles, respectively. The refractive index of the sample was evaluated using a THz transmission measurement at n≈2.1 over the relevant frequency range. A scan of the entire sample surface area ($120\times 120\;{\mathrm{mm}}^{2}$) was performed in a measurement time of ≈5 min (scanning velocity $200\;\frac{\mathrm{mm}}{s}$, line distance 0.25mm). The 3D dataset obtained with the algorithm defined in Section 2.4 represents the reflectivity of the sample as a function of the lateral surface coordinates x,y and the optical depth Λ in the sample. For example, Figure 7b shows the top and bottom surface of the GFRP plate. The periodic structure of the top surface is caused by the 2 × 2 twill weaving pattern of the glass fiber plies as described in Section 2.1. Due to the linear polarization of the electric field of the THz pulse [35], the reflectivity is dependent on the lateral fiber orientation, leading to the observed periodically varying reflectivity. Note that not only the top ply but also each individual glass fiber ply below the surface produces a similar pattern. However, the influence of scattering and the plies located above may overshadow the pattern until it becomes more and more blurred, such as in the bottom layer in Figure 7b.

A slice along the xΛ plane, commonly referred to as B-scan in 3D imaging, is shown in Figure 7c. This slice image was measured along the blue dashed line inserted in Figure 7a across the six aluminum inlays in increasing depths from 0.75 mm up to 2.63 mm. One can clearly identify the top and bottom surface of the GFRP plate (marked in green) and the individual inlays marked in red.

Additional slices along the xy plane in two different depths (0.75 mm and 1.13 mm), commonly referred to as C-scans, are shown in Figure 7d. Listed from left to right, each C-scan includes a 10×10 aluminum, 2×2 aluminum, 10×10 PTFE, 5×5 PTFE, and a 2×2 PTFE inlay (units in mm). Note that the right side of the C-scan, which includes the three PTFE inlays, is amplified such that the colormap is not distorted by the much stronger metal inlay reflection amplitudes. The detected inlays are marked in red again. Although all inlays can be detected successfully; the reflection caused by the 2×2 PTFE inlay in a depth of 1.13 mm is only marginally stronger than other background reflections caused by the surrounding glass fiber plies of the GFRP plate. Therefore, it is probably difficult to identify this inlay type in >1.13 mm depth reliably in practice.

From these results, it can be inferred that defects inducing a strong change in optical properties such as metallic inclusions will be detectable in this type of GFRP sample down to 3 mm or more, while defects with optical properties closer to the GFRP such as the Teflon inlays will only be detectable within the first couple of millimeters. Air gaps present an intermediate case between these two types of defects. We therefore expect to be able to detect delaminations within the first 1 or 2 mm of GFRP.

### 3.2. Delaminations in GFRP Drilled Holes

The two drilled GFRP plates shown in Figure 1 are scanned in a measurement time of ≈ 30min (scan velocity $200\;\frac{\mathrm{mm}}{s}$, line distance of 0.25mm). In order to validate the results obtained with THz reflection imaging, 36 drilled holes (out of 165 holes in total) are scanned using the high-resolution X-ray CT system described in Section 2.5. To reduce the measurement time and costs, and to reach an acceptable resolution with XCT, the samples have to be cut into smaller parts (150 mm × 15 mm). Four samples containing 9 holes, shown in Figure 1 by red dashed boxes, are scanned. The raw X-ray CT projection data are reconstructed mathematically into a 3D image using a filtered back-projection algorithm [36]. In total, the 36 3D X-ray CT images represent 28 GB of measurement data, while the 3D THz datasets obtained with the signal processing algorithm discussed in Section 2.4 represent 40 GB of data for 165 holes.

In the following, pairs of C-scans at equal depths from the 3D THz and 3D X-ray CT datasets are compared. To make the comparison possible, the geometrical depth in the THz images is calculated based on the knowledge of the sample thickness and the localization of the top and bottom surfaces. The resolution of the THz images is necessarily lower than for X-CT images due to the much longer wavelength of the THz radiation.

Two pairs of C-scans through the same hole at different depths are shown in Figure 8. The first pair of C-scans in Figure 8a,b is measured at a depth of 0.3 mm below the surface and shows a significant defect in the upper region of the hole (marked in blue). In the X-ray image, the defect can be recognized as a coherent region of low intensity. On the other hand, the THz image shows only a distinct peak in the left region of the dashed box. This can be explained by the fact that THz imaging reveals high intensities at regions with abrupt changes in the refractive index (i.e., an air–GFRP material transition). Therefore, only boundaries are visible in the THz images, whereas the X-ray images show the actual density of the volume. The discrepancy between XT and THz images could also come from a tilt of the delamination plane with regards to the imaging plane, which would impact the THz image. A second potential defect can be observed on the right side of the hole. An inhomogeneity can be found in both images, but it is not clear whether this feature is caused by the natural woven structure of the GFRP plate or by a real defect. Additional C-scans through the same hole at a depth of 0.6 mm are shown in Figure 8c,d. According to the X-ray image, the defect becomes narrower until it eventually vanishes. In the THz image, two peaks in the right half of the defect become more pronounced, whereas the strong peak in the left side vanishes. This indicates the presence of a material interface in the right half of the dashed box. These defects are not immediately visible to the naked eye when looking into the holes.

Figure 9 shows two pairs of C-scans obtained from two additional holes (hole #2 and #3). In the X-ray image in Figure 9a, a small defect is marked in blue. A significant peak at the same position is identified in the THz image (see Figure 9b). The C-scan pair measured around hole #3 in Figure 9c,d shows two defects in the X-ray image. Since the left defect appears to be geometrically larger, it is expected to cause a stronger THz reflection, which is confirmed by the corresponding THz image.

The first pair of C-scans in Figure 10a,b shows an identified delamination near the surface around the hole. In Figure 10c,d, a defect adjacent to the hole is found in both the X-ray and THz C-scans. Additionally, one further potential defect is visible in the top right corner of the blue frame in the X-ray image. Once again, it is not fully clear whether this is a real defect or caused by the natural structure of the material, and the defect is not visible to the naked eye.

## 4. Discussion

This work presents a novel method for the detection of defects in drilled GFRP using a newly developed complete, compact, fast, and accurate THz reflection imaging system. A complete data-processing algorithm to generate 3D THz images is developed. At present, this algorithm is applied in a MATLAB post-processing routine; however, parts of the algorithm are already implemented in the real-time imaging software (e.g., the process of interpolation). In a future step, the entire algorithm could be implemented directly in the imaging software, enabling real-time 3D data visualization. The real-time visualization capabilities of the imaging software currently include the possibility to show B- and C-scans.

First measurements on a GFRP sample including different types of artificial defects (aluminum or PTFE inlays) of different sizes at different depths allowed to estimate the resolution and sensitivity of the system. It was found that aluminum inlays with a size of $10\times 10\;{\mathrm{mm}}^{2}$ can be detected reliably up to a depth of at least 2.63 mm. Additionally, several PTFE inlays (with sizes $10^{2},5^{2},2^{2}\;{\mathrm{mm}}^{2}$) at depths of 0.75 mm and 1.13 mm were detected. While the larger inlays were detected reliably, the reliable detection of the smaller inlays (<$2\times 2\;{\mathrm{mm}}^{2}$) in depths >1.13 mm is most certainly not possible in practice. For this special material, the inhomogeneous background reflections caused by its woven structure are limiting the reliability for smaller and/or deeper defects.

The performance of the system was evaluated by comparing volumetric THz scans of drilled holes to volumetric X-ray CT scans. Matching defects in X-ray CT and THz imaging were found and discussed around five drilled holes. The depth of the identified defects varied between 0.3 mm and 0.6 mm; therefore, THz imaging was demonstrated to be a viable alternative to X-ray imaging in this depth range. For significantly larger depths, a better choice of THz emitters and detectors operating at lower frequencies may offer improvement. Furthermore, a new generation of emitters with significantly higher output power [37,38] may increase the penetration depth. The resolution is limited by the long wavelength as well as the optical elements used. Decreasing the wavelength would not lead to significant improvement, as higher frequencies are strongly absorbed by the material. Therefore, the resolution can only be marginally improved with the current system.

One challenging aspect of THz imaging in GFRP parts made of 2 × 2 twill prepregs is the fact that the naturally inhomogeneous structure of the sample produces THz reflections even in the absence of delaminations. Although it was found that defects in the above-mentioned depth range generated significantly stronger reflections near the boundary of the hole, it was sometimes not clear whether a detected feature corresponded to a defect or a natural reflection. One way to address this challenge would be to perform a THz measurement before drilling and another one after drilling. This way, a differential measurement highlighting the structural changes induced in the material by drilling could be calculated.

In conclusion, the demonstrated system offers several practical advantages compared to X-ray or ultrasound imaging, such as the simplicity, the high accuracy near the surface and the use of non-ionizing radiation. These advantages can be highly beneficial especially in industrial environments for improving the quality control of produced GFRP parts.

## Figures and Tables

**Figure 1 sensors-25-00851-f001:**
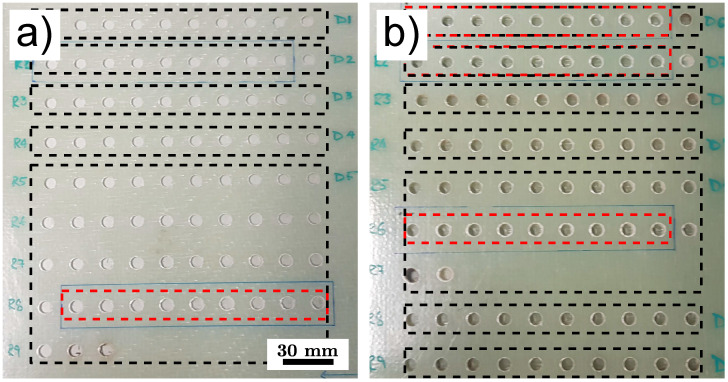
Drilled GFRP plates: (**a**) 3 mm thick plate, (**b**) 6 mm thick plate. Dashed black boxes: holes drilled with the same drill bit sequentially from the top left corner to bottom right corner. Dashed red boxes: sections cut out afterwards to perform XCT measurements.

**Figure 2 sensors-25-00851-f002:**
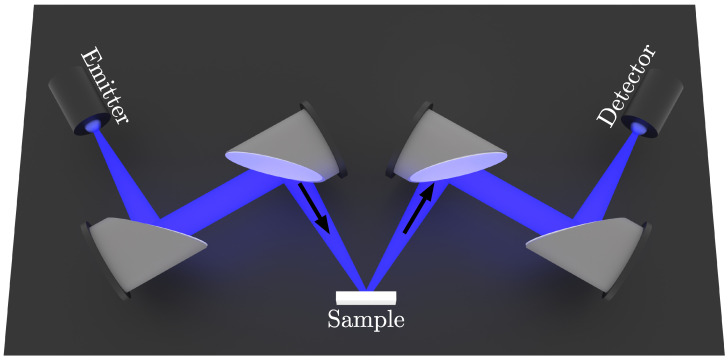
THz reflection imaging scheme with 2 PCAs (emitter and detector) and 4 off-axis parabolic mirror guiding and focusing the THz beam onto the sample.

**Figure 3 sensors-25-00851-f003:**
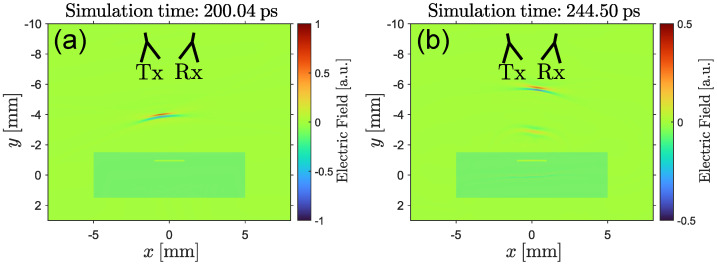
Interaction of THz radiation with a delamination: snapshots of a simulation of a THz pulse propagating towards a GFRP plate (dark area) with a delamination (small slit). (**a**) THz pulse traveling from the emitter (Tx) towards the GFRP plate (dark area). (**b**) Incoming THz pulse partially reflected at the top surface of the GFRP plate and the delamination. The image shows both partial reflections propagating towards the detector (Rx) after coupling out of the sample again.

**Figure 4 sensors-25-00851-f004:**
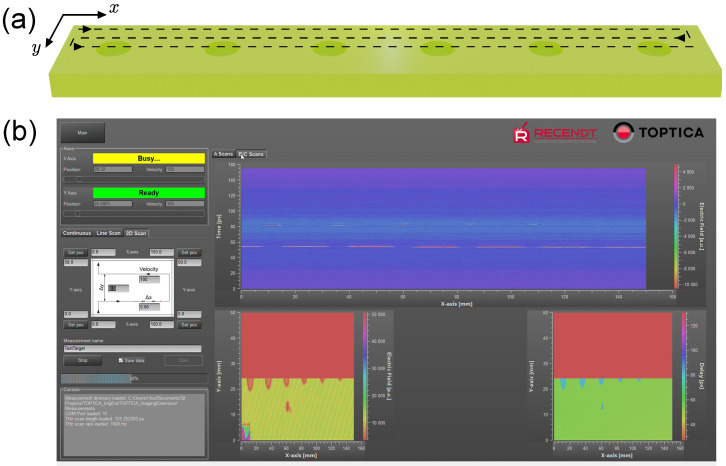
Scanning trajectory and real-time visualization. (**a**) Schematic of the GFRP plate’s drilling process and scanning trajectory in the x−y plane. Scan performed continuously. (**b**) Screenshot of the imaging software interface during scanning, showing the live visualization of the measurement data. The left side of the interface allows to set the scanning parameters and control the axis movement. On the right side, the current B-scan and the progress of two C-scans showing the pulse amplitude and time delay.

**Figure 5 sensors-25-00851-f005:**
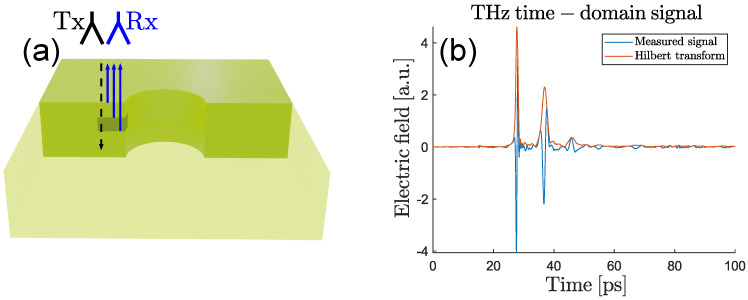
Optical detection of a delamination by reflection imaging. (**a**) Schematic image of a drilled hole in a GFRP plate including a delamination (dark area). The THz radiation propagates from the emitter (Tx) into the sample (black dashed arrow) and generates partial reflections (blue arrows) at interfaces that propagate back towards the detector (Rx). (**b**) Raw time-domain THz measurement signal (blue) and its Hilbert transform (orange).

**Figure 6 sensors-25-00851-f006:**
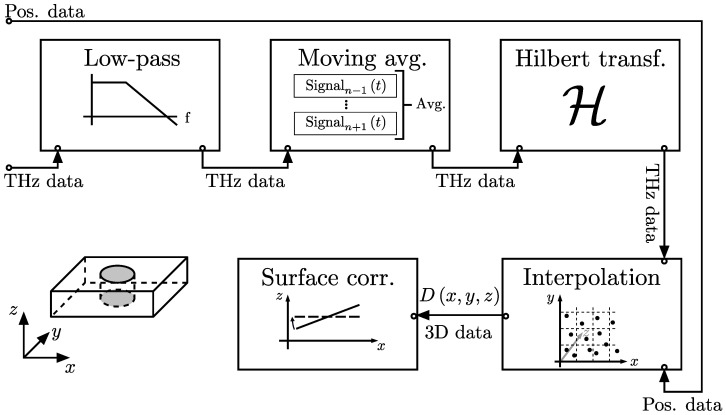
Signal processing algorithm. The raw data produced by the imaging system (i.e., measurement positions and THz time-domain signals) are processed by a low-pass and moving average filter. The 3D volumetric image is calculated by applying the Hilbert transform to the time-domain signal, interpolating the measurement data on a 3D spatial grid. Additionally, the dataset is rotated in space such that the sample surface is aligned in the x-y plane.

**Figure 7 sensors-25-00851-f007:**
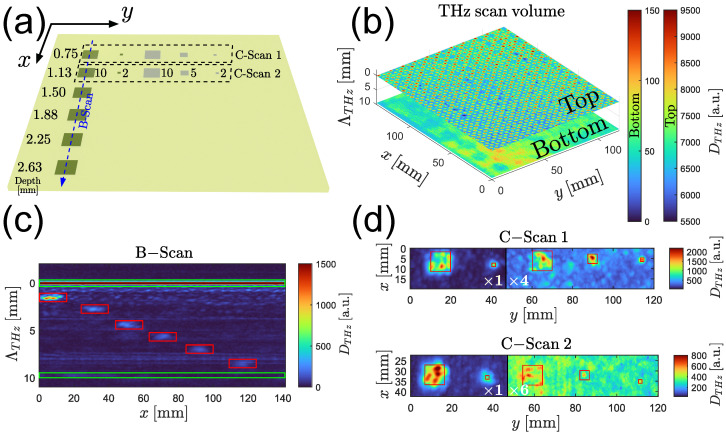
The 3D THz imaging of artificial inlays in a GFRP plate. (**a**) Sketch of the sample showing the lateral location and depth of the aluminum inlays (dark areas) and PTFE inlays (gray areas). Location and orientation of the slices shown in (**c**,**d**) are indicated by the dashed lines/boxes. (**b**) Top and bottom surfaces extracted from the measured 3D THz dataset. (**c**) B-scan along the blue dashed line in (**a**), THz reflections from top and bottom surfaces (green boxes) and reflections from the individual aluminum inlays (red boxes). (**d**) C-scans extracted at a depth of 0.75 mm and 1.13 mm from dashed regions in (**a**), location of the individual detected inlays (red boxes).

**Figure 8 sensors-25-00851-f008:**
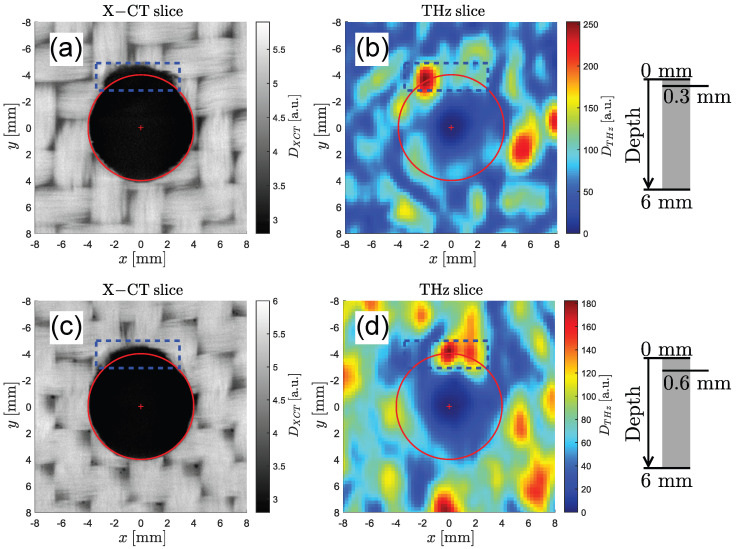
Drilled hole #1. (**a**,**b**) Pair of C-scans (X-ray left and THz right) at a depth of 0.3 mm. A clearly visible defect in the top area of the hole is highlighted (dashed box) in both the X-ray and THz images. (**c**,**d**) Additional pair of C-scans at a depth of 0.6 mm. The X-ray image shows the same but slightly narrower defect, whereas the THz image indicates the presence of a material interface in the right half of the dashed region.

**Figure 9 sensors-25-00851-f009:**
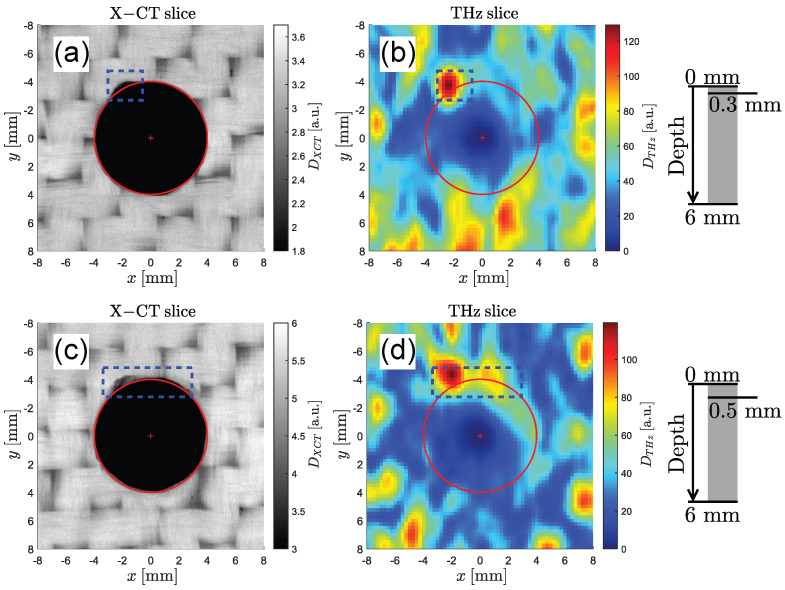
Drilled holes #2 and #3. (**a**,**b**) X-ray CT scan of hole #2 showing a delamination at a depth of 0.3 mm in the top left corner (blue frame). The same feature is also identified as a bright spot in the THz scan at the same depth. (**c**,**d**) A delamination in hole #3 is found at a depth of 0.5 mm (blue frame). Similarly, two large reflections are found in the same region in the THz scan.

**Figure 10 sensors-25-00851-f010:**
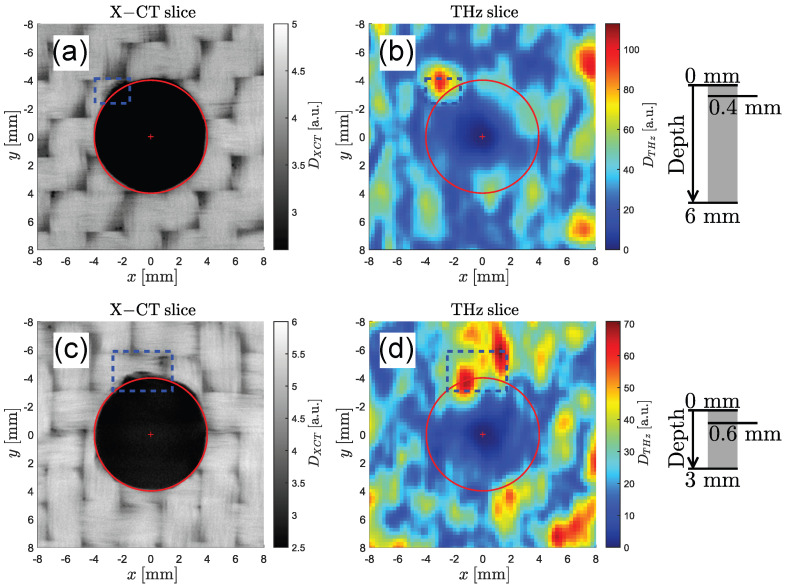
Drilled holes #4 and #5. (**a**) X-ray image of hole #4 0.4 mm below the surface of a 6 mm GFRP plate and (**b**) corresponding THz image. (**c**) Hole #5 drilled into a 3 mm GFRP plate and imaged using X-ray CT 0.6 mm below the surface. (**d**) Corresponding THz image showing identical defect features around the top region of the hole.

## Data Availability

The raw data supporting the conclusions of this article will be made available by the authors on request.

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
