# Peer review of "Fast Terahertz Reflection Imaging for In-Line Detection of Delaminations in Glass Fiber-Reinforced Polymers"

_sensors, 2025, doi:10.3390/s25030851_

Round 1

Reviewer 1 Report

Comments and Suggestions for Authors

Interesting paper on valuable technique. However, description of prior state of art research is not sufficient. I did not find an explanation of novelty of this paper compared with 2017 paper "Nondestructive evaluation of GFRP composite including multi-delamination using THz spectroscopy and imaging" https://doi.org/10.1016/j.compstruct.2017.11.012. Actually, 400 results in Google Scholar for query "GFRP thz tds" and none are discussed.

Reference [3] leads to https://doi.org/10.1016/j.compscitech.2006.10.009 where such a statement is made:
"For instance, in the aircraft industry drilling associated delamination accounts for 60% of all rejections during final assembly of an aircraft."

I do not agree that it is a good reference to substantiate this "delamination damages during final assembly was as high as 60%" number.

Photos in Figure 1 are blurry and not aid visual analysis.

Author Response

We thank the reviewer for their thoughtful review. Please see the attachment for the answers to the comments.

Reviewer 2 Report

Comments and Suggestions for Authors

The paper reported a fast Terahertz reflection imaging for in-line detection of delaminations in glass-fiber reinforced polymers. It’s an interesting work. However, upon careful examination of the manuscript, I found that their results did not convince me to recommend acceptance. Additional data also needs to be supplemented to provide support and evidence for their findings and claimed breakthroughs. Here are detailed technical suggestions and comments.

1. The innovation of this paper is not outstanding enough. Compared with the existing THz-TDS scanning system, what is the greatest contribution of this paper?

2. The paper claims that "This system, based on the THz-TDS system TeraFlash smart developed by TOPTICA Photonics AG, exceeds the performance of current available solutions in terms of measurement speed ". If this is indeed the case, it is suggested to give a specific numerical value, and at the same time give a comparison with the data of the same type of instruments in this field.

3. The scanning results of 3D imaging are not good, with low resolution. Is there any improvement plan?

4. Compared with X-ray CT, the imaging effect of terahertz tomography is poor, whether the scanning platform is leveled and whether the influence of diffraction is taken into account.

Author Response

(The authors gave the same response as above.)

Reviewer 3 Report

Comments and Suggestions for Authors

This manuscript demonstrates a THz 3D imaging system for investigating defects in glass-fiber reinforced polymers. Basic principle of the system is given, and experimental results of the system for drilled GFRP samples compared with X-ray measurements are provided. The experimental results are reliable. I think some issues need to be solved before further consideration.

1. As a research paper, the novelty of the work is not well highlighted. In my opinion, both the system setup and the algorithm for reconstructing the image seem not new. The authors should give more clearly what is the advancement of this system.

2. In the discussion section, the authors claim that one advantage of the system is measurement speed: ‘This system, based on the THz-TDS system TeraFlash smart developed by TOPTICA Photonics AG, exceeds the performance of current available solutions in terms of measurement speed and is currently in the progress of being commercialized in cooperation with TOPTICA Photonics AG.' What is the basis of this conclusion? The authors should provide a comparison table listing current products or solutions with speeds. In addition, the authors also explain why this system has higher speed than others.

3. As a research paper, some key information is not provided in this manuscript. such as

(1) To realize 3D imaging, Eq. (1) is the basic principle, in which the refractive index of the material is a key parameter. However, the refractive index of any material is not given in the manuscript. What are their values in various THz frequencies? How were they obtained? 

(2) It is quite strange that the wavelength or frequency of the THz wave is not mentioned in the manuscript because it is well known that the image resolution is dependent on the wavelength. What is the wavelength for Fig. 7-10?

4. The reason that the authors use Hilbert transform is not given. What is the advantage of this method?

5. In the explanation of Fig. 8, the authors mention 'This can be explained by the fact that THz imaging reveals high intensities at regions with abrupt changes of the refractive index (i.e. an air-GFRP material transition).'(Line 296-297) What is the physical mechanism of this conclusion?

Author Response

(The authors gave the same response as above.)

Round 2

Reviewer 1 Report

Comments and Suggestions for Authors

The changes make it a much more compelling paper. Also, I would add 1.6 kHz sampling rate in this statement on line 62: 

"Its key advantage is its measurement speed, much higher than that of systems using a mechanical delay stage (around 100 Hz) or even the fast ASOPS (Asynchronous Optical Sampling) systems (up to 1 kHz)."

Also, as mentioned before statement on line 29 is not well supported by [3]:

"Delamination damages during final assembly were responsible for as much as 60% of part rejections [3]."

Overall, the work introduces novel technique for faster damage inspection in multi layer composites.

Author Response

Comment 1: Also, I would add 1.6 kHz sampling rate in the statement on line 62

Response 1: We added the 1.6 kHz sampling rate to the statement. Thank you for the suggestion.

Comment 2:  The statement on line 29 is not well supported by [3]: "Delamination damages during final assembly were responsible for as much as 60% of part rejections [3].
Response 2: This statement can be found in a number of publications on the subject of delamination damage in composites. When tracking down the original publication, we found it must originally come from a conference proceeeding paper, which we unfortunately could not find online. We thus decided to leave out this sentence entirely.

Reviewer 2 Report

Comments and Suggestions for Authors

In my opinion, the revised manuscript has made better improvements.

Author Response

We thank again the reviewer for their time spent reviewing our manuscript.

Reviewer 3 Report

Comments and Suggestions for Authors

The authors have answered all my questions. I suggest this manuscript be accepted.

Author Response

(The authors gave the same response as above.)
